# Multilevel theorizing in health communication: Integrating the Risk Perception Attitude (RPA) framework and the Theory of Normative Social Behavior (TNSB)

**Hagere Yilma** [1]*, **Rajiv N. Rimal**[2], **Manoj Parida**[3]

**1** Department of Health Sciences, Boston University College of Health & Rehabilitation Sciences: Sargent College, Boston, Massachusetts, United States of America, **2** Department of Health, Behavior and Society Johns Hopkins University Bloomberg School of Public Health, Baltimore, Maryland, United States of America, **3** D-COR (Development Corner) Consulting Pvt. Ltd., Odisha, India

* hy@bu.edu

**Data Availability Statement:** The data has been shared in a public repository; the DOI for the data is: 10.6084/m9.figshare.16713277.

## Abstract

Research testing the risk perception attitudes (RPA) framework has demonstrated that efficacy can moderate the effect of risk perceptions on behavior. This effect of efficacy has also been seen at the social-level through tests of the theory of normative social behavior (TNSB). We tested if efficacy could bridge normative factors at a social-level and risk perception at an individual-level. Data for this study come from the Reduction in Anemia through Normative Innovations (RANI) project's baseline survey in Odisha, India. We used hierarchical regressions to analyze interactions between predictors at various levels and efficacy to predict behavioral intention. Efficacy beliefs moderated the effect of injunctive norms ($\beta = 0.07$, $p < 0.01$), collective norms ($\beta = 0.06$, $p < 0.01$), and risk perception ($\beta = 0.04$, $p < 0.01$) on intentions. This study provides preliminary evidence for a multilevel theoretical framework.

## Introduction

Risk perception is an important concept in a number of health behavior theories, including the protection motivation theory [1], the health belief model [2], and the extended parallel process model [3]. These theories propose that a heightened sense of vulnerability to a disease motivates people to take action to avert the threat. Empirical findings, however, are not as straightforward. In some instances, risk perception is positively associated with health behaviors [4, 5], while in others it can be negatively or not associated at all with behaviors [6, 7]. A meta-analysis on vaccination behaviors found an overall effect size of $r = .24$ for susceptibility (perception that one is vulnerable to a disease) and $r = .16$ for severity (perception about the seriousness of the disease) [6]. Another meta-analysis, focusing on colorectal screening behavior, found an effect size of $z = .13$, with higher-quality studies showing smaller effects than lower-quality studies [8].

**Funding:** This study was funded by the Bill and Melinda Gates Foundation (OPP1182519). The authors of this paper have no conflicts of interest to report. The grant was warded to RR as the principle investigator. https://www.gatesfoundation.org/ The funders had no role in study design, data collection and analysis, decision to publish, or preparation of the manuscript.

**Competing interests:** The authors have declared that no competing interests exist.

Many researchers have identified reasons why risk perception may not map onto behaviors. Some authors have noted that a relationship between risk perceptions and behaviors is better captured through longitudinal than cross-sectional data [6, 9, 10]. Others have attributed the inconsistency in the relationship between risk perceptions and behaviors to poor measures of underlying constructs [6, 11, 12]. Researchers have also noted that risk perceptions are only relevant for health protective behaviors where there is some sense of personal loss [13, 14].

Other researchers have noted that the relationship between risk perceptions and behaviors has to take into account people's efficacy beliefs–that, in order to translate risk-induced motivation into actual action, people must feel efficacious [15–17]. Rimal and Juon (2010) reported that risk perceptions were positively associated with breast self-exams when efficacy beliefs were strong, but not when efficacy beliefs were weak [18]. These and other similar results indicate that, in order for risk perceptions to be translated into protective behaviors, people must feel confident in their ability to perform the behavior of interest and believe that the behavior will yield desirable outcomes [19, 20]. This relationship between risk perception and efficacy beliefs is outlined in the risk perception attitudes (RPA) framework, which posits that, when efficacy beliefs are strong, risk perceptions are translated into behavior change more readily, but this process is severely hampered when efficacy beliefs are weak [21, 22]. Thus, the ability of heightened risk perceptions to propel behavior change appears to be contingent upon the strength of extant efficacy beliefs.

The importance of efficacy beliefs, however, extends beyond its relationship with risk perception. Efficacy can shape the meaning ascribed to outcomes, such that a relapse is viewed as evidence of one's inability when one's efficacy is weak, and as evidence of not having expended adequate effort when one's efficacy is strong [23]. Efficacy also guides choices people make, often averting challenges when efficacy is weak and taking on more difficult tasks when it is strong [24–27].

Beyond the influences at the individual level, efficacy beliefs also serve to link social beliefs with outcomes. Tests of the theory of normative social behavior (TNSB) have shown that descriptive norms (perceptions about the extent to which peer groups or similar others engage in a behavior) are stronger predictors of behavior intentions when self-efficacy is strong than when it is weak [28, 29]. Recent conceptualizations of TNSB indicate that injunctive norms (perceptions about social approval) and collective norms (true prevalence of the behavior) may also be moderated by certain factors in the same manner as descriptive norms [30–32].

Given this important moderating role of efficacy beliefs, in this paper we propose and test the idea that efficacy beliefs act as the bridge that links individual-level psychosocial drivers of behavior change, on the one hand, with social-level factors, on the other. In particular, we attempt to link the social-level relationships outlined in the TNSB with individual-level interactions proposed in the RPA framework. Furthermore, risk perceptions have not yet been incorporated into the TNSB, and it appears that efficacy beliefs provide a conceptual mechanism for doing so. Efficacy beliefs, after all, are related to both individual and social environments, as strong efficacy beliefs are characterized by confidence in ability to engage in a behavior, despite barriers, including social barriers.

## Hypotheses

We begin with the idea that people often act in accordance with their perceptions of others' actions (i.e., based on descriptive norms) and that feeling efficacious to act is a key component in doing so. After all, if a behavior is thought to be difficult to undertake or if one's actions are perceived to be inconsequential, how many others are engaging in that behavior is likely to have little impact on one's intentions to engage in the behavior [33, 34]. This idea is also

supported by findings from tests of the TNSB, which have investigated the moderating role of efficacy in the relationship between descriptive norms and behaviors. The finding is that the relationship between descriptive norms and behaviors is heightened when efficacy is strong and attenuated when efficacy is weak [28, 29]. As our first hypothesis, we test this idea in a cultural context (in India) and behavioral domain (taking iron folic acid tablets) in which the TNSB has not been tested previously.

**Hypothesis 1 (H1).**   The relationship between descriptive norms and behavioral intentions will be moderated by efficacy beliefs, such that this relationship will be stronger when efficacy beliefs are strong than when efficacy beliefs are weak.

We also have reasons to believe efficacy beliefs will moderate the influence of injunctive norms on behaviors. Consider, for example, a situation in which one's efficacy beliefs are strong. In this scenario, when pressures to conform (i.e., injunctive norms) are also strong, one has both the ability and the motivation to act, resulting in a higher likelihood of action [35]. When efficacy beliefs are strong but pressures to conform are absent, individuals' ability remains high but they are less motivated to act, resulting in an attenuated likelihood of action. In the converse scenario, when efficacy beliefs are weak–thus signaling to the individual that they are unable to enact the behavior–pressures to conform are likely to have only minimal effects. Hence, when efficacy is low, we expect the relationship between injunctive norms and behaviors to be rather weak. In the literature, we are unable to find studies that have tested this proposition. One study found that efficacy beliefs moderated the role of peer feedback on behavior [36]. In this study, peer feedback was conceptualized as the number of supportive or opposing social media messages about the self-protective behavior–not quite (though somewhat close to) injunctive norms. We thus propose:

**Hypothesis 2 (H2).**   The relationship between injunctive norms and behavioral intentions will be moderated by efficacy beliefs, such that this relationship will be stronger when efficacy beliefs are strong than when efficacy beliefs are weak.

The literature on the relationship between collective norms and behavior has also yet to be unpacked in this manner. In using the term collective norms, we are adopting the definition proposed by Sedlander and Rimal (2019)–the aggregate of behaviors within a community or peer group [37]. Unlike descriptive and injunctive norms, which are based on people's perceptions about others' behaviors and attitudes, respectively, collective norms constitute the sum total of behaviors individuals are exposed to in their social environments. Using this conceptualization, a woman living in a village in which most others take iron tablets to prevent anemia, for example, would be characterized as having high collective norms with regard to this behavior. The impact of this norm, which can occur independently from perceptions, can be manifest in a variety of ways: perhaps she sees others engage in the behavior, finds discarded tablets in the trash can, or observes their sale in the local pharmacy. This is also akin to the idea of social exposure, articulated by Mead, Rimal, Ferrence, and Cohen (2014), to mean the sum total of all instances in which one is exposed to the enactment of a particular behavior in one's midst [38].

The literature on collective norms is underdeveloped, but the few studies that have investigated collective norms have found a positive relationship between collective norms and behavior [39, 40]. We also expect self-efficacy to facilitate collective norms' influence on behaviors. When most others in one's social environment are enacting a behavior and one has high levels of efficacy to follow suit, fewer barriers exist to dampen the propensity to act. Conversely, if one perceives low efficacy to act and most others in one's social environment are also not acting, then few motivations exist to act. If, however, one feels highly efficacious to act but others in one's environment are not acting, there are fewer motivations to act, despite higher levels of ability. Lastly, if one feels unable to act, but lives in an environment where many others are

engaging in the behavior, we suspect that other's actions serve to highlight one's inability, which would lead to inaction. But another scenario is also possible: others' actions (particularly if the others are similar to oneself) could propel one to experiment with the behavior, despite having low efficacy. It appears we do not yet have sufficient evidence to form a hypothesis. Instead, we raise the following research question:

**Research Question 1 (RQ 1).**   What is the nature of the interaction between efficacy beliefs and collective norms in their impact on behavioral intentions?

We expect the relationship between risk perception, efficacy, and behavior intentions to be similar to that found in previous tests of the RPA framework [15–22]. However, the studies yield inconsistent findings around the effect of risk perception at low levels of efficacy. In some cases, risk perceptions were not a significant predictor of behavioral intentions at low levels of efficacy, but in other cases, the effect of risk perception was still significant but attenuated at low levels of efficacy. The latter pattern was more common in studies where health information seeking was the outcome, while the former was a common finding in studies where other self-protective behaviors were the outcome [41]. In this study, our outcome of interest (as noted below) falls into this category of self-protective behaviors. Thus, we expect the effect or risk perception on behavioral intentions to be significant at high, but not at low, levels of efficacy.

**Hypothesis 3 (H3).**   The relationship between risk perceptions and behavioral intentions will be moderated by efficacy beliefs, such that this relationship will be stronger when efficacy beliefs are strong than when efficacy beliefs are weak

The full conceptual model can be found in Fig 1.

## Materials and methods

### Study design

Data for this study come from the Reductions in Anemia through Normative Innovations (RANI) Project [42]. The RANI project uses a clustered randomized control trial (cRCT) to

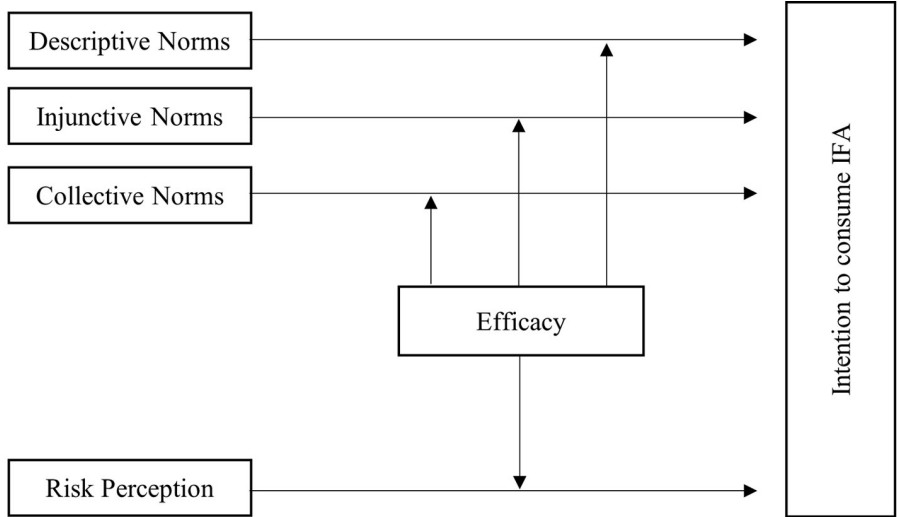

**Fig 1. Conceptual model.** This figure depicts the proposed conceptual model-an integration of TNSB and RPA in which efficacy acts as a moderator to the normative predictors outlined in TNSB and risk perception.

evaluate the ability of a norms-based iron and folic acid (IFA) supplementation promotion intervention in Odisha, India. The RANI project is being run in two blocks (a block is analogous to a county in the United States) within the Angul district in the state of Odisha. Because this paper is based on data from the baseline assessment of the project, we will not dwell on the randomized design, except to note that data come from 81 villages in both treatment and control arms.

The cRCT is registered in the Clinical Trial Registry of India (CTRI; CTRI/2018/10/016186) and has received approval from Indian Council for Medical Research's (ICMR's) Health Ministry's Screening Committee (HMSC), which, together with Sigma–a firm in New Delhi–serve as the local Institutional Review Boards. A review board from George Washington University in the United States approved the study. We obtained written informed consent from all participants immediately prior to data collection. As a part of the consent process, data collectors read allowed the written informed consent document approved by the IRB to each study participant. Participants indicated consent by signing the document. If a participant was unable to sign their own name, data collectors signed on their behalf. If a participant was below the age of 18, data collectors obtained informed assent from the participant, as well as permission from their parent (indicated by signature). Participants retained a copy of their informed consent document.

## Participants

The RANI Project randomly selected households using a number generator from a household listing exercise conducted by data collectors. From the selected home, one woman of reproductive age (between 15 and 49 years old) was chosen (randomly if more than one woman was eligible in the same home). Although our sample consisted of 4,110 participants, this paper restricts the sample to those who were not pregnant at the time of survey ($n$ = 3,913). We do so because the primary dependent variable–taking iron and folic acid tablets–has been heavily promoted among pregnant women by the Government of India. Pregnant women are also enrolled into the medical system, where physicians or community workers provide iron and folic acid for free. This is not the case among non-pregnant women, whose decisions to take iron and folic acid are made on their own, likely based on their own psychosocial factors, their social networks, and their family (i.e., not by the medical system). The demographic profile of participants included in our analysis is shown in Table 1.

## Measures

**Descriptive norms.** Descriptive norms were operationalized as the average proportion of similar others believed by participants to regularly take IFA supplements. Investigations of TNSB have found that similarity perceptions can influence the effect of norms, particularly as it relates to efficacy beliefs [43]. In this study, similarity was determined by pregnancy status. To this effect, if the respondent was pregnant, she was asked "What proportion of pregnant women in your community take IFA regularly?" If the respondent was not pregnant, she was asked "What proportion of non-pregnant women in your community take IFA regularly?" (Only non-pregnant women are included in this paper). Responses were recorded on a 4-point scale where 0 corresponded to "none," 1 corresponded to "some," 2 corresponded to "about half," 3 corresponded to "most," and 4 corresponded to "all" ($M$ = 0.25, $SD$ = 0.51).

**Injunctive norms.** Injunctive norms were operationalized as the extent to which women believed their influential referent groups expected them to take iron and folic acid supplements. Analogous to the assessment of descriptive norms, injunctive norms were also measured with respect to pregnancy status. Respondents who were not pregnant were asked the

**Table 1. Demographic profile of participants.**

|  | *M* | *SD* | *%* |
|---|---|---|---|
| Age | 30.59 | 8.83 |  |
| Current IFA consumption | 0.16[a] | 1.18 |  |
| Education |  |  |  |
| None |  |  | 18.6 |
| Up to class 5 |  |  | 24.7 |
| Up to class 12 |  |  | 53.2 |
| > class 12 |  |  | 3.5 |
| Married |  |  | 79.2 |
| Religion-Hindu |  |  | 99.8 |
| # of Children |  |  |  |
| None |  |  | 22.7 |
| 1 |  |  | 19.8 |
| 2 |  |  | 35.1 |
| 3 |  |  | 14.7 |
| 4 |  |  | 5.4 |
| 5 or more |  |  | 2.3 |
| Ever asked for IFA |  |  | 5.7 |

IFA = iron and folic acid

Note: [a] Current IFA consumption was calculated by averaging the number of iron and folic tablets consumed within the past 7 days.

extent to which they agreed with two statements: "Your mother-in-law (or mother, if unmarried) thinks you should take iron and folic acid tablets regularly, even if you are not pregnant" and "Your husband (or father, if not pregnant) thinks you should take iron and folic acid tablets regularly, even if you are pregnant." Husbands and mothers-in-law were chosen as the referent groups for measurement because of our formative work that highlighted their importance. Responses were recorded on a 5-point Likert scale and averaged into a single measure for injunctive norms ($\alpha$ = 0.45; $M$ = 2.50, $SD$ = 1.16). The poor reliability indicated by the low Cronbach's alpha is of concern, and we bring this up as a limitation in the paper.

**Collective norms.** Collective norms were operationalized as the true prevalence of intention to take iron and folic acid within a cluster, calculated by computing the non-self-mean of intentions within each village ($M$ = 3.72, $SD$ = 0.39). This method of calculating collective norms by aggregating behaviors within a community or selected peer group has been used in other studies [37, 39].

**Efficacy.** Similar to other studies, efficacy beliefs comprised self-efficacy (one's confidence to enact a behavior) and response efficacy (belief that one's behaviors will result in desired outcomes) [22]. Self-efficacy was measured by asking participants the extent to which they agreed with the statements "You can take iron and folic acid every week when you are not pregnant," "You believe that you could easily take iron and folic acid," "You can take iron and folic acid even if your husband/father does not want you to do so," and "You can take iron and folic acid even if your mother/mother-in-law does not want you to do so". All responses were recorded on a 5-point Likert scale ($\alpha$ = 0.87).

Similarly, response efficacy was measured by asking participants the extent to which they agreed with two statements "Taking iron and folic acid regularly will make you feel stronger" and "Taking iron and folic acid every day can help prevent fatigue and dizziness during

pregnancy." Responses were recorded on 5-point Likert scales ($\alpha = 0.78$). All six items were averaged into a single efficacy index ($\alpha = 0.84$; $M = 3.88$, $SD = 0.93$).

**Risk perception.**   Risk perception was operationalized as the perceived level of susceptibility and severity. Perceived susceptibility was conceptualized as the belief that one could become anemic. Pilot testing of measurement items indicated that it was difficult for our population to answer scaled questions, so a dichotomous measure was used. Participants were asked "do you believe that you will become anemic in the coming year?" Participants who reported "yes" or that they were already anemic were categorized as feeling susceptible (score of 1) and those who responded "no" were categorized as not feeling susceptible (score of 0). Perceived severity was conceptualized as the extent to which one believed that becoming anemic was negative. It was measured by asking participants the extent to which they agreed with the statement "If you become anemic, it would affect your health in a negative way." In order to keep this consistent with perceived susceptibility, responses were coded on 5-point Likert scale and converted into a dichotomous category where those who strongly agreed were given a score of 1 (72.2%) and those who disagreed (strongly or at all), were neutral, or agreed (not strongly) were given a score of 0. Responses to both perceived susceptibility and perceived severity were added together. The resulting index for risk perception ranged from 0 to 2, where 0 indicated having poor perceptions of both susceptibility and severity, 1 indicated having strong perceptions of just one of these risk dimensions, and 2 indicated having strong perceptions of both susceptibility and severity ($M = 1.30$, $SD = 0.69$).

**Intention to take iron and folic acid.**   Intention to take iron and folic acid was conceptualized as the extent to which women expressed that they would take the tablets in the future. Studies have shown that intentions are a reliable predictor of subsequent behaviors [44]. Respondents were asked the extent to which they agreed with the following statements: "If you were to get pregnant in the future, you will take iron and folic acid every-day," "You will take iron and folic acid once a week in the future, even if you are not pregnant," "If you are not pregnant you will take iron and folic acid every week even if your husband/male member in your community does not think it is a good idea," and "If you are not pregnant you will take iron and folic acid every week even if your mother-in-law/woman in your community does not think it is a good idea." All responses were recorded on a 5-point scale and averaged into an index for intentions ($\alpha = 0.83$; $M = 3.71$, $SD = 1.09$). The scale was negatively skewed, so the final outcome variable was exponentially transformed through an iterative process using Tukey's ladder of powers until normality was achieved.

## Statistical analysis

Descriptive statistics were used to understand the demographic and psychosocial profile of the sample. Hierarchical regression equations were used to test study hypotheses. In the first step of the analysis, we entered demographic control variables, including age and education. We then controlled for previous consumption behaviors because studies have shown that previous engagement in behaviors can influence both risk appraisals and future behaviors [9, 45, 46].

In the second step, all main-effects of interest were entered into the model. This included descriptive norms, injunctive norms, collective norms, risk perception, and efficacy beliefs.

The third and final step of the models included the interaction term between efficacy and the other variables. Each interaction term was entered in the model, tested for significance, and then removed before the next interaction term was entered (to reduce multicollinearity). If an interaction term was found to be significant, the nature of the interaction was plotted using Aiken and West (1991) recommendations [47]. This was done by first centering the variables around their mean and standardizing them (with a mean of zero and standard deviation

of one). Interactions were investigated at three levels of the moderator (in this case, efficacy): at 1 *SD* above the mean (high efficacy), at the mean (medium efficacy), and at 1 *SD* below the mean (low efficacy).

# Results

## Description of sample

The average age of the sample was 30.59 years (*SD* = 8.83). The majority of the sample (79.2 percent) was married at baseline and Hindu (99.8 percent). Most of the sample (57.4 percent) also had two or more children. Additionally, most of the sample (97.3 percent) was not currently taking iron and folic acid, and only 7.1 percent had ever asked for iron and folic acid. The full demographic profile of participants can be found in Table 1.

## Hypothesis tests

Results of the tests of our primary hypotheses, conducted through hierarchal linear regressions, are shown in Table 2. In the first step of the model, we entered age, education, and current iron and folic acid consumption as control variables. The fit statistics (adjusted R-squared) indicated that these variables accounted for 1.2 percent of the total variance in intentions. In the second step, we included the psychosocial variables of interest as main-effects. Interactions were entered one at a time in subsequent steps. Table 2 shows beta coefficients of the final model without the interactions (i.e., with only all the main-effects in the model).

The linear regressions showed that age, education, and current iron and folic acid consumption were not significantly associated with intention to take iron and folic acid in the future when the psychosocial factors of interest were controlled for.

**Table 2. Effects of norms, risk perceptions, and efficacy on intentions to consume iron and folic acid from regression equations.**

| Predictors | $\beta^a$ | $\Delta R^2$ |
|---|---|---|
| Step 1: Controls | | .011 |
| Age | -.01 | |
| Education | .007 | |
| Current IFA consumption | .004 | |
| Step 2: Main effects[b] | | .683 |
| Descriptive norms | .02* | |
| Injunctive norms | .08*** | |
| Collective norms | .06*** | |
| Risk perception | .04*** | |
| Efficacy | .77*** | |
| Step 3A: Descriptive norms x efficacy | .01 | n.s. |
| Step 3B: Injunctive norms x efficacy | .07*** | .004 |
| Step 3C: Collective norms x efficacy | .05*** | .002 |
| Step 3D: Risk perception x efficacy | .04*** | .001 |

Notes.

[a]Standardized betas from regression equations after entering all the main-effects into the model.

[b]Standardized betas for the main-effects and interaction terms listed in steps 2 and 3 were calculated by first centering the variables around a mean of zero and standardizing them.

* p < .05

** p < .01

*** p < .001

In Step 2 of the model, we entered the psychosocial factors of interest as main-effects. The addition of these predictors added an additional 68.2 percent of variance in intentions. These variables included the norms outlined in the TNSB: descriptive norms, injunctive norms, and collective norms. All three norms were positively associated with intentions. The more participants believed that similar others were taking iron and folic acid (i.e., strong descriptive norms), the more likely they were to take IFA in the future ($\beta = 0.02$, $p < 0.05$). Similarly, injunctive norms ($\beta = 0.08$, $p < 0.001$) and collective norms ($\beta = 0.06$, $p < 0.001$) were also associated with intention. The second step of the model also contained risk perception and efficacy beliefs as main-effects. Both risk perception ($\beta = 0.04$, $p < 0.001$) and efficacy beliefs ($\beta = 0.77$, $p < 0.001$) were positively associated with intentions.

We entered the interaction terms corresponding to our study hypotheses and research questions into the model one at time, resulting in four different final models (step 3A –step 3D). The addition of these interaction terms explained an additional one to four percent of the variance in intention; the overall models explained up to 69.8 percent of the variance in intention.

In Step 3A, we entered the interaction between descriptive norms and efficacy into the model. This interaction term was not significant, and hence H1 was not supported. In step 3B, the interaction term between descriptive norms and efficacy was removed, and the interaction term between injunctive norms and efficacy was added. This interaction term was significant, indicating that the effect of injunctive norms was significantly moderated by efficacy beliefs ($\beta = 0.07$, $p < 0.001$; Fig 2). The relationship between injunctive norms and intentions was

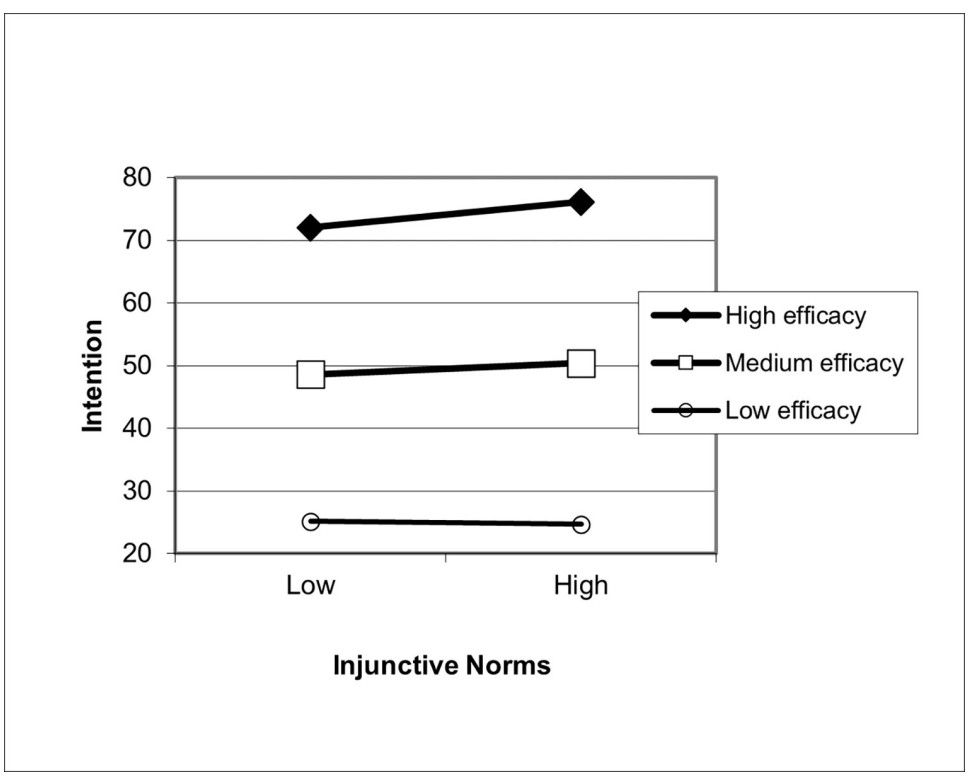

**Fig 2. Relationship between injunctive norms and intentions moderated by efficacy beliefs.** The interaction between injunctive norms and efficacy in predicting intentions is plotted here. Injunctive norms are plotted on the x-axis; intentions is plotted on the y-axis. The best fitting line for intention regressed onto injunctive norms is shown by varying levels of efficacy. High efficacy is defined as 1 standard deviation above the mean, medium efficacy is defined as the mean, and low self-efficacy is defined as one standard deviation below the mean.

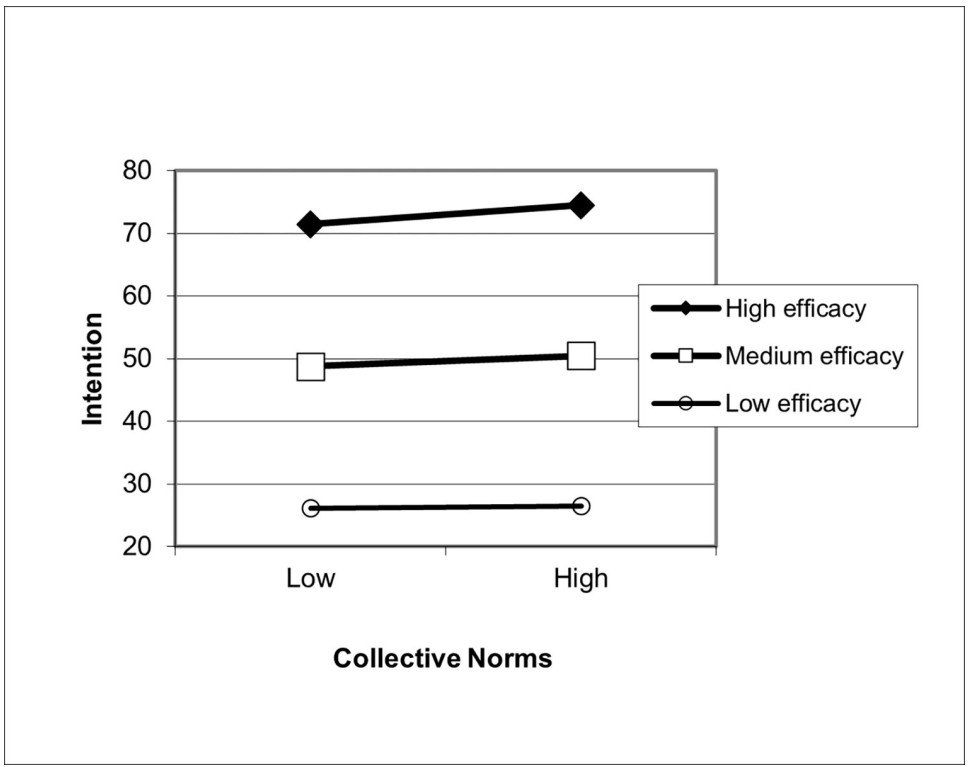

**Fig 3. Relationship between collective norms and intentions moderated by efficacy beliefs.** The interaction between collective norms and efficacy in predicting intentions is plotted here. Collective norms are plotted on the x-axis; intentions is plotted on the y-axis. The best fitting line for intention regressed onto collective norms is shown by varying levels of efficacy. High efficacy is defined as 1 standard deviation above the mean, medium efficacy is defined as the mean, and low self-efficacy is defined as one standard deviation below the mean.

significant only when efficacy beliefs were high or average, but not when they were low. Thus, we found support for H2.

In Step 3C, this interaction term was replaced by an interaction term between collective norms and efficacy. This interaction term was also significant, indicating that the association between collective norms and intentions was significantly moderated by efficacy beliefs ($\beta$ = 0.06, $p$ < 0.001; Fig 3). The relationship between collective norms and intentions was significant at high and medium levels of efficacy, but not at low levels of efficacy. We thus found preliminary evidence to answer RQ1.

Lastly, in Step 3D, the interaction term between risk perception and efficacy beliefs was entered into the model. The interaction term was significant ($\beta$ = 0.04, $p$ < 0.001; Fig 4). The relationship between risk perceptions and intentions was significant when efficacy beliefs were high or medium, but not when efficacy beliefs were low. Thus, we found support for H3.

## Discussion

This study investigated the extent to which efficacy beliefs would act as a bridge between social-level factors (as operationalized through social norms) and individual-level factors (in this case, risk perception). Our findings highlight the important role of efficacy beliefs in moderating the effect of both normative influences and risk perception.

All three norms (descriptive, injunctive, and collective) outlined in the modified version of the TNSB were significantly associated with behavioral intentions (although descriptive

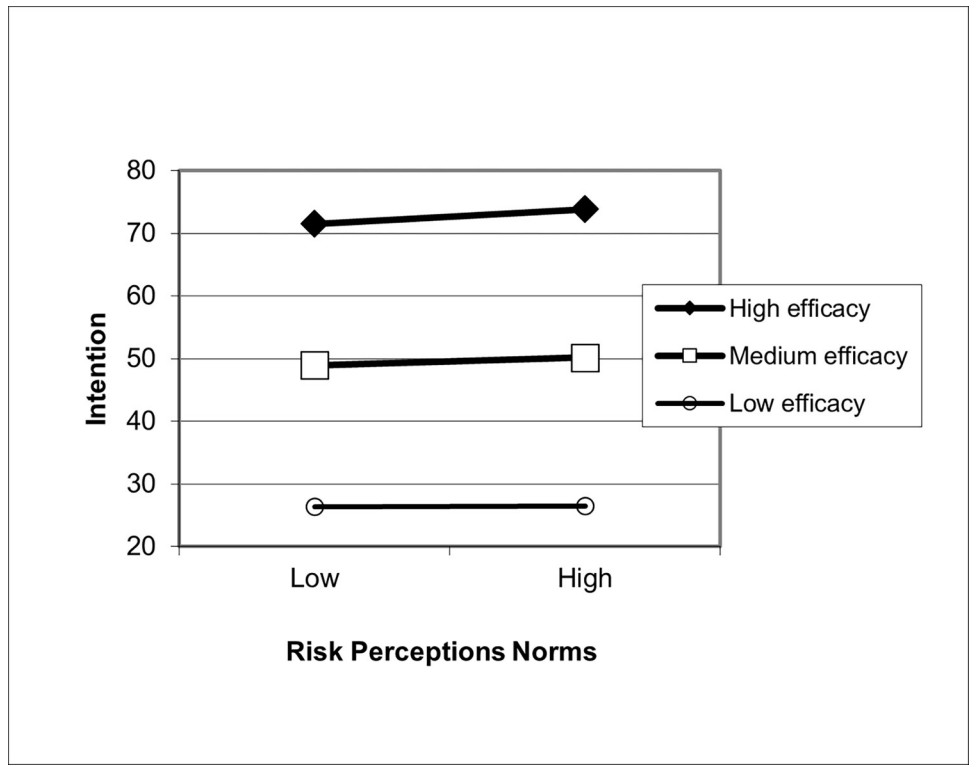

**Fig 4. Relationship between risk perceptions and intentions moderated by efficacy beliefs.** The interaction between risk perception and efficacy in predicting intentions is plotted here. Risk perceptions are plotted on the x-axis; intentions is plotted on the y-axis. The best fitting line for intention regressed onto risk perceptions is shown by varying levels of efficacy. High efficacy is defined as 1 standard deviation above the mean, medium efficacy is defined as the mean, and low self-efficacy is defined as one standard deviation below the mean.

norms' contribution was less so, as we discuss below) [30]. Believing that many similar others are taking iron folic acid, feeling pressures to do so, and coming from environments in which others are also engaging in the behavior were associated with one's intention to engage in the same behavior. Similarly, risk perception was also found to be significantly associated with intentions. We should note that these main-effects on future intentions were found after controlling for individuals' own prior behaviors.

Perhaps one of the most significant findings reported in this paper was the strength of the association between efficacy beliefs and intentions. In the multivariate model, the standardized beta coefficient of this relationship was high, β = .77 ($p < .001$), signifying its robust association with intentions, even after controlling for other factors in the model. The main-effect of efficacy beliefs is also apparent in the graphical display of our findings (Figs 2 and 4).

Apart from the strength of this main-effect, we also found evidence of the interaction effects of efficacy beliefs. At the individual level, self-efficacy boosted the association between injunctive norms and intentions and between collective norms and intentions. Those who felt efficacious were more likely to react positively to pressures they perceived to conform. Similarly, efficacious individuals were also more likely to translate their heightened risk perceptions into self-protective behavioral intentions.

Our findings show that efficacy beliefs can set the conditions for factors at both the individual- and social-level. At the individual-level, risk perceptions follow the same boundary conditions; risk perceptions are propelled into behavioral intention when the individual feels strong

efficacy. In the absence of efficacy, people's risk perceptions did not translate into behavioral intentions. At the social-level, believing that others are engaging in a protective behavior, perceiving strong levels of social approval for that behavior, and existing in social contexts where that behavior is prevalent are likely to translate into action when the individual has confidence to enact the behavior [28, 29].

Contrary to prior research, however, we did not find descriptive norms to have much impact on behavioral intentions. Even though its main-effect was statistically significant, the small effect size ($\beta$ = .02) tells us that the large sample size was likely responsible for this finding. Given the plethora of findings from across health domains showing the association between descriptive norms and behaviors [48, 49], that we did not find similar results in this paper leads us to believe our measure was likely not robust enough. Indeed, it was assessed with just a single measure (what proportion of other, similar women take iron folic acid), which likely introduced a significant level of error in the data. We also know from formative research that thinking probabilistically and proportionately, as is required to answer questions about descriptive norms, does not come naturally to many people, especially in lower-education settings [42]. That we asked such a "difficult" question and did so through just a single item were likely the primary reasons why we failed to see associations with descriptive norms. Indeed, we ran additional bivariate analyses and found that, among the three norms, descriptive norms had the weakest zero-order correlation with intentions ($r$ = .15, $p$ < .001); (injunctive norms: $r$ = .40, $p$ < .002; collective norms: $r$ = .31, $p$ < .001). This further leads us to believe that the lack of association in the multivariate model was not just because of shared variance between descriptive norms and other predictors (though there is likely some of that), and that measurement error was likely key.

The pattern of interaction effects is also worth noting (Figs 2 and 4. In both norms-related interactions with efficacy beliefs (injunctive norms and collective norms), the relationship between norms and efficacy beliefs were significant when efficacy beliefs were strong. This relationship further weakened as efficacy beliefs grew weaker, such that, at low levels of efficacy beliefs, the relationship between norms and intentions were no longer significant. Exactly the same pattern was found when risk perception was the independent variable: Perceptions of susceptibility and severity can only translate into intentions when efficacy is strong, supporting previous tests of the RPA framework [15, 16, 18–22].

Overall, our results indicate that shifting social norms within communities and improving people's risk perceptions are not enough to propel behavior if individuals do not believe that they can engage in that behavior and that the behavior will result in desired outcomes.

The integration of the RPA framework and TNSB extends the purview of both of these studies. The investigation of social norms in health communication research has shifted from demonstrating the propensity of people to change behaviors (because they perceive that others around them are changing) to trying to uncover *how* and *when* norms can influence behavior [30, 50]. Much of the literature in this respect has been focused on the effect of descriptive norms, while collective and injunctive norms have been largely ignored as primary predictors. To our knowledge, this is the first time the influence of efficacy has been investigated as a moderator between these two forms of norms and behaviors.

## Limitations

Three limitations are worth noting here. First, this study used cross-sectional data to predict behavior intentions. As noted in previous research, the effect of risk perception is best captured through longitudinal data [9, 10]. Previous research has also shown that the relationship between risk perception and intention is often bidirectional and they indicate that risk

perceptions are poorer after one engages in protective behaviors [9, 45, 46]. Our results show a positive correlation between risk perception and behaviors; if the outcome (i.e., behaviors) were to precede the predictor (i.e., risk perceptions), then we would expect to see a negative correlation between the two. This does, however, warrant further research.

Direction of the relationship between efficacy and behaviors is another significant issue. Indeed, the extraordinarily high correlations between efficacy beliefs and intentions leads us to suspect that many participants were inferring efficacy from their own intentions. Those who believed they would engage in the behavior may well have then come to believe that doing so was within their repertoire of abilities and that doing so would be effective. This is definitely worth exploring through longitudinal data.

Lastly, intention to consume IFA, rather than actual consumption, is used as the primary outcome in this study. This was in part done to minimize obscurity in causality, as the measurement of behaviors in cross-sectional data would require us to compare previous behaviors with current psychosocial predictors. While intentions have been found to predict a large portion of the variance in behaviors, there is still some variance unexplained [51]. The meta-analysis conducted by Cooke et al., 2016 found that intention is a better predictor for older populations, as younger populations are less likely to translate their intention into behavior [51]. Additionally, other structural barriers related to the supply and distribution of iron folic acid may prevent intentions from being translated into action. Nevertheless, intention is still consistently shown to be an adequate proxy of behaviors [52].

## Conclusion

Informed primarily by social cognitive theory, enhancing efficacy to enact health behaviors has been a well-known strategy in the health campaign literature for many years [25]. Indeed, Bandura (1977) has explicated many reasons as to why efficacy is such a powerful predictor of behavior [24]. What has not been explored in the literature is the extent to which efficacy beliefs serve as a bridge to connect individual-level motivations with social- and environment-level factors that either promote or inhibit the translations of those motivations into actions. Pan and McLeod (1991) first articulated micro and macro level linkages in mass communication research, and a special issue of *Communication Research* was devoted to this idea [53]. Even now, few guidelines exist on how we develop theoretical constructs that span multiple levels. Efficacy beliefs appear to offer one way of further exploring cross-level linkages, and this paper has provided an initial roadmap for doing so.

## Author Contributions

**Conceptualization:** Hagere Yilma.

**Data curation:** Hagere Yilma, Rajiv N. Rimal.

**Formal analysis:** Hagere Yilma, Rajiv N. Rimal.

**Funding acquisition:** Rajiv N. Rimal.

**Investigation:** Hagere Yilma, Rajiv N. Rimal.

**Methodology:** Hagere Yilma, Rajiv N. Rimal.

**Project administration:** Manoj Parida.

**Supervision:** Rajiv N. Rimal, Manoj Parida.

**Writing – original draft:** Hagere Yilma, Rajiv N. Rimal, Manoj Parida.

**Writing – review & editing:** Hagere Yilma, Rajiv N. Rimal, Manoj Parida.

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
