## [Decision Letter · Decision Letter 0]

19 Apr 2022

PONE-D-21-35351Multilevel theorizing in health communication: integrating the Risk Perception Attitude (RPA) framework and the Theory of Normative Social Behavior (TNSB)PLOS ONE

Dear Dr. Hagere Yilma,

Thank you for submitting your manuscript to PLOS ONE. After careful consideration, we feel that it has merit but does not fully meet PLOS ONE’s publication criteria as it currently stands. Therefore, we invite you to submit a revised version of the manuscript that addresses the points raised during the review process.

We look forward to receiving your revised manuscript.

Kind regards,

Lei Shi

Academic Editor

PLOS ONE

3. Please ensure you have included the registration number for the clinical trial referenced in the manuscript.

“This study was funded by the Bill and Melinda Gates Foundation (OPP1182519). The authors of this paper have no conflicts of interest to report.”

“This study was funded by the Bill and Melinda Gates Foundation (OPP1182519). The authors of this paper have no conflicts of interest to report. The grant was warded to RR as the principle investigator.

https://www.gatesfoundation.org/

“No authors have competing interests”

Reviewers' comments:

Reviewer's Responses to Questions

**Comments to the Author**

1. Is the manuscript technically sound, and do the data support the conclusions?

Reviewer #1: Yes

Reviewer #2: Yes

2. Has the statistical analysis been performed appropriately and rigorously? 

Reviewer #1: Yes

Reviewer #2: Yes

3. Have the authors made all data underlying the findings in their manuscript fully available?

Reviewer #1: Yes

Reviewer #2: Yes

4. Is the manuscript presented in an intelligible fashion and written in standard English?

Reviewer #1: Yes

Reviewer #2: Yes

5. Review Comments to the Author

Reviewer #1: This is a well written manuscript and the statistical analyses approach in a stepwise fashion is routine for this type of question. The sample size is certainly adequate and the power needed to address the hypotheses may be adequate, although the investigators are counting on the original randomized trial to accommodate this concern. Admittedly the study is cross sectional.

The limitations are well explained and the conclusions follow from the analyses provided. The investigators used hierarchical linear equations to analyze interactions between predictors at various levels and efficacy to predict behavioral intention. The moderators incorporated into the model are certainly reasonable. There is no mention of any fit statistics examining the appropriateness of the linear models unless this is already discussed in previous work and should be referenced if such is the case.

Reviewer #2: This paper resorts the concept of efficacy to bring together normative factors at the social-level, specifically including descriptive norms, injunctive norms and collective norms, and risk perception at the individual-level, and analyzes data from the RANI project through hierarchical linear equations, and then examines the role of a series of predictors and efficacy on the intention to consume IFA. The main findings are the strength of the association between efficacy beliefs and intentions and the interaction effect involving efficacy beliefs.

The literature review of this paper is well prepared, the hypotheses are natural and reasonable, and the conclusions are generally credible. It is an honor to read this paper, but there are still a few suggestions before publication:

1. It is suggested that the authors supplement the hierarchical linear equations used in this paper at appropriate place, and further enrich the summary statistics on measures;

2. It is suggested that the authors provide further explanations for “low reliability” and “exponentially iterative process” in Measures.

3. As the authors mention the relationship between efficacy and behaviors in Limitations, it is also recommended that they explore the self-efficacy and response efficacy separately;

4. The dependent variable in this paper is the intention to consume IFA, not the actuals. If permitted, it is recommended that the authors continue to track the behavioral performance of participants to better supplement and evaluate the conclusions involved in this paper.

6. PLOS authors have the option to publish the peer review history of their article (what does this mean?). If published, this will include your full peer review and any attached files.

Reviewer #1: No

Reviewer #2: No

---

## [Author Response · Author response to Decision Letter 0]

8 Jun 2022

Dear Reviewers ,

Thank you for taking the time to thoroughly review our manuscript PONE-D-21-35351

Multilevel theorizing in health communication: integrating the Risk Perception Attitude (RPA) framework and the Theory of Normative Social Behavior (TNSB). I am delighted to share our response to each editorial and reviewer comment below, indicated in bold blue font within our attached letter (Response to reviewers. doc). 

Reviewer #1: This is a well written manuscript and the statistical analyses approach in a stepwise fashion is routine for this type of question. The sample size is certainly adequate and the power needed to address the hypotheses may be adequate, although the investigators are counting on the original randomized trial to accommodate this concern. Admittedly the study is cross sectional.

The limitations are well explained and the conclusions follow from the analyses provided. The investigators used hierarchical linear equations to analyze interactions between predictors at various levels and efficacy to predict behavioral intention. The moderators incorporated into the model are certainly reasonable. There is no mention of any fit statistics examining the appropriateness of the linear models unless this is already discussed in previous work and should be referenced if such is the case.

Thank you for your thoughtful review and summary of the strengths and limitations of our study. We report the adjusted R-square as the fit statistic for the model with the main effects, as well as changes in adjusted R-square with the addition of each moderator (page 15-16). We have now edited the manuscript so that this clearer.

Reviewer #2: This paper resorts the concept of efficacy to bring together normative factors at the social-level, specifically including descriptive norms, injunctive norms and collective norms, and risk perception at the individual-level, and analyzes data from the RANI project through hierarchical linear equations, and then examines the role of a series of predictors and efficacy on the intention to consume IFA. The main findings are the strength of the association between efficacy beliefs and intentions and the interaction effect involving efficacy beliefs.

The literature review of this paper is well prepared, the hypotheses are natural and reasonable, and the conclusions are generally credible. It is an honor to read this paper, but there are still a few suggestions before publication:

1. It is suggested that the authors supplement the hierarchical linear equations used in this paper at appropriate place, and further enrich the summary statistics on measures;

Thank you for your thoughtful and careful review of our manuscript. We have not included hierarchical linear equations in our manuscript because we only present models with variables measured at a singular level (the individual level). Rather, the “hierarchical regressions” referred to in this paper is in reference to the step-wise fashion in which the interaction terms were included into the model. 

2. It is suggested that the authors provide further explanations for “low reliability” and “exponentially iterative process” in Measures.

Thank you for this note. We have edited the manuscript to reflect that we are referring to the low Cronbach’s alpha when we use claim the measure has low reliability, and that the iterative process used was Tukey’s ladder of powers. 

3. As the authors mention the relationship between efficacy and behaviors in Limitations, it is also recommended that they explore the self-efficacy and response efficacy separately

Thank you for this note. We have looked at efficacy as separate variables and have not found a difference in findings between the two. However, after much consideration, we have decided to report efficacy as a combination of self-efficacy and response-efficacy because the RPA framework indicates that it is the combination between the two that has the power to strengthen or attenuate the influence of risk perception on behavior intentions.

4. The dependent variable in this paper is the intention to consume IFA, not the actuals. If permitted, it is recommended that the authors continue to track the behavioral performance of participants to better supplement and evaluate the conclusions involved in this paper.

We appreciate this comment and agree that behavior is important to track. We are delighted to share that the behavioral impact of the RANI Project will be reported in the intervention’s final impact paper. However, in terms of this particular paper, we have chosen to look at intention rather than behavior because 1) we wanted to stay true to the theories used; the TNSB and the RPA framework posit that risk perception, efficacy, and social norms predict intentions, which then subsequently impact behavior and 2) intention and behaviors are measured 6 to 12 months a part in subsequent rounds of the RANI Project’s data collection, therefore we do not report associations between intentions and behavior here due to the temporal distance between the two measures. However, we have added a statement under our description of the measurement of intentions that supports the association between intentions and behaviors.

---

## [Decision Letter · Decision Letter 1]

8 Jul 2022

Multilevel theorizing in health communication: integrating the Risk Perception Attitude (RPA) framework and the Theory of Normative Social Behavior (TNSB)

PONE-D-21-35351R1

Dear Dr. Hagere Yilma,

We’re pleased to inform you that your manuscript has been judged scientifically suitable for publication and will be formally accepted for publication once it meets all outstanding technical requirements.

Kind regards,

Lei Shi

Academic Editor

PLOS ONE

Additional Editor Comments (optional):

Reviewers' comments:

Reviewer's Responses to Questions

**Comments to the Author**

1. If the authors have adequately addressed your comments raised in a previous round of review and you feel that this manuscript is now acceptable for publication, you may indicate that here to bypass the “Comments to the Author” section, enter your conflict of interest statement in the “Confidential to Editor” section, and submit your "Accept" recommendation.

Reviewer #2: All comments have been addressed

2. Is the manuscript technically sound, and do the data support the conclusions?

Reviewer #2: Yes

3. Has the statistical analysis been performed appropriately and rigorously? 

Reviewer #2: Yes

4. Have the authors made all data underlying the findings in their manuscript fully available?

Reviewer #2: Yes

5. Is the manuscript presented in an intelligible fashion and written in standard English?

Reviewer #2: Yes

6. Review Comments to the Author

Reviewer #2: (No Response)

7. PLOS authors have the option to publish the peer review history of their article (what does this mean?). If published, this will include your full peer review and any attached files.

Reviewer #2: No

---

## [Editor Report · Acceptance letter]

14 Jul 2022

PONE-D-21-35351R1 

Multilevel theorizing in health communication: integrating the Risk Perception Attitude (RPA) framework and the Theory of Normative Social Behavior (TNSB) 

Dear Dr. Yilma:

I'm pleased to inform you that your manuscript has been deemed suitable for publication in PLOS ONE. Congratulations! Your manuscript is now with our production department. 

Kind regards, 

on behalf of

Dr. Lei Shi 

Academic Editor

PLOS ONE